# How Accurate and Precise Can We Measure the Posture and the Energy Expenditure Component of Sedentary Behaviour with One Sensor?

**DOI:** 10.3390/ijerph18115782

**Published:** 2021-05-27

**Authors:** Roman P. Kuster, Wilhelmus J. A. Grooten, Victoria Blom, Daniel Baumgartner, Maria Hagströmer, Örjan Ekblom

**Affiliations:** 1Division of Physiotherapy, Department of Neurobiology, Care Sciences and Society, Karolinska Institutet, 141 83 Stockholm, Sweden; wim.grooten@ki.se (W.J.A.G.); maria.hagstromer@ki.se (M.H.); 2IMES Institute of Mechanical Systems, School of Engineering, ZHAW Zurich University of Applied Sciences, 8401 Winterthur, Switzerland; daniel.baumgartner@zhaw.ch; 3Women’s Health and Allied Health Professionals Theme, Medical Unit Occupational Therapy and Physiotherapy, Karolinska University Hospital, 171 77 Stockholm, Sweden; 4Department of Physical Activity and Health, The Swedish School of Sport and Health Sciences, 114 86 Stockholm, Sweden; victoria.blom@gih.se (V.B.); orjan.ekblom@gih.se (Ö.E.); 5Division of Insurance Medicine, Department of Clinical Neuroscience, Karolinska Institutet, 171 77 Stockholm, Sweden; 6Academic Primary Health Care Center, Region Stockholm, 104 31 Stockholm, Sweden

**Keywords:** ActiGraph, activPAL, calibration, free-living behaviour, machine learning, objective measurement, office worker, physical activity, Posture and Physical Activity Index (POPAI), validation

## Abstract

Sedentary behaviour is an emergent public health topic, but there is still no method to simultaneously measure both components of sedentary behaviour—posture and energy expenditure—with one sensor. This study investigated the accuracy and precision of measuring sedentary time when combining the proprietary processing of a posture sensor (activPAL) with a new energy expenditure algorithm and the proprietary processing of a movement sensor (ActiGraph) with a published posture algorithm. One hundred office workers wore both sensors for an average of 7 days. The activPAL algorithm development used 38 and the subsequent independent method comparison 62 participants. The single sensor sedentary estimates were compared with Bland–Atman statistics to the Posture and Physical Activity Index, a combined measurement with both sensors. All single-sensor methods overestimated sedentary time. However, adding the algorithms reduced the overestimation from 129 to 21 (activPAL) and from 84 to 7 min a day (ActiGraph), with far narrower 95% limits of agreements. Thus, combining the proprietary data with the algorithms is an easy way to increase the accuracy and precision of the single sensor sedentary estimates and leads to sedentary estimates that are more precise at the individual level than those of the proprietary processing are at the group level.

## 1. Introduction

Sedentary behaviour is an emerging public health topic. In November 2020, the World Health Organization (WHO) released a new guideline recommending adults limit their sedentary time because higher amounts of sedentary behaviour are associated with several detrimental health effects, such as all-cause, cardiovascular, and cancer mortality, as well as the incidences of cardiovascular disease, type 2 diabetes, and cancer [1]. The WHO adopted the sedentary behaviour definition of the Sedentary Behavior Research Network, defining sedentary behaviour as “any waking behaviour characterised by an energy expenditure of 1.5 METs (Metabolic Equivalents) or lower while sitting, reclining or lying” [1,2]. This definition includes a posture (sitting, reclining, or lying) and an energy expenditure component (≤1.5 METs). However, the evidence synthesis on which the WHO relied is almost exclusively based on movement sensors classifying energy expenditure [3]. In fact, Katzmarzyk and colleagues concluded in 2019 when updating the evidence synthesis for the American guidelines that there is a pressing need to develop sensor-based methods to simultaneously assess the two components of sedentary behaviour that can be applied in surveillance and research settings to properly quantify sedentary time [4]. The need was recently repeated while updating the WHO guidelines [5]. To date, however, there is still a lack of methods to measure both components of sedentary behaviour simultaneously with a single sensor applicable to surveillance and research settings.

For a couple of years, it has been well established that body-worn accelerometers should be the method of choice to measure sedentary time [6]. Depending on the placement and way of data processing, they can be divided into two groups: posture sensors and movement sensors [7]. The most frequently used posture sensor is probably the activPAL (PAL Technologies Ltd., Glasgow, SCO), for which the time classified as sitting is taken as an estimate of sedentary behaviour [8,9]. The most frequently used movement sensor is probably the ActiGraph (ActiGraph LCC, Pensacola, FL, USA) worn at the waist, for which the minutes with fewer than 100 counts (a proprietary acceleration unit) on the vertical sensor axis are typically taken as an estimate of sedentary behaviour [3,10]. Thus, the activPAL’s sedentary estimate is exclusively based on posture (i.e., sitting), while the ActiGraph’s sedentary estimate is exclusively based on energy expenditure (i.e., ≤1.5 METs). For both sensor types, efforts have been made to develop new algorithms to classify the other component, i.e., energy expenditure for posture sensors [11,12] and posture for movement sensors [13,14,15]. However, thus far, these algorithms have not been combined with the proprietary data processing of the sensors to classify posture and energy expenditure simultaneously.

We recently presented a simultaneous measurement of both components of sedentary behaviour: The Posture and Physical Activity Index (POPAI) [16]. POPAI combines the proprietary data processing of a thigh-worn activPAL with the proprietary data processing of a waist-worn ActiGraph. Each activPAL sitting event is classified on a minute-by-minute level into inactive (≤1.5 METs) and active (>1.5 METs) using the ActiGraph counts. Compliant with the definition, only the inactive sitting time is considered sedentary. The comparison to the single sensors revealed that they substantially overestimate sedentary time by 30.3% (activPAL) and 22.5% (ActiGraph). This observation confirmed the results previously observed for other posture and movement sensor combinations [17,18]. In our study, the overestimation of the activPAL could be perfectly explained with active sitting (r^2^ = 1.0), while the overestimation of the ActiGraph could be almost perfectly explained with inactive standing (r^2^ = 0.92) [16]. However, the most serious limitation of POPAI is the need for two sensors, significantly limiting its suitability for application in surveillance and research settings. Therefore, the present study examined the accuracy and precision of measuring both components of sedentary behaviour with a single sensor, a thigh-worn activPAL (proprietary data with and without a new energy expenditure algorithm) and a waist-worn ActiGraph (proprietary data with and without a published posture algorithm).

## 2. Materials and Methods

To measure both components of sedentary behaviour with one sensor, the proprietary posture classification of the activPAL was combined with an energy expenditure algorithm to activPAL+, while the proprietary energy expenditure classification of the ActiGraph was combined with a posture algorithm to ActiGraph+. The development of the activPAL algorithm is presented here, while the development of the ActiGraph algorithm is presented in [15]. Both algorithm developments used the same data and development procedure. All single sensor sedentary estimates (activPAL, activPAL+, ActiGraph, and ActiGraph+) were then compared to POPAI to analyse how accurately (group level) and precisely (individual level) the single sensors measure sedentary behaviour compliant with its definition.

### 2.1. Participants

The study used the same 100 office workers as in [16]. The same 38 participants already used for the development of the ActiGraph algorithm were used to develop the activPAL algorithm (development sample), and the remaining 62 participants were used for the subsequent independent method comparison (comparison sample, Figure 1). The development sample consisted of 25 men and 13 women with an average age of 42.3 ± 8.4 years and an average body mass of 71.2 ± 10.2 kg. The comparison sample consisted of 36 men and 26 women with an average age of 39.6 ± 9.4 years and an average body mass of 71.8 ± 14.1 kg.

### 2.2. Data Recording

All participants were recorded within the Brain-Health-Study investigating the association of physical activity and sedentary behaviour patterns to cognition, mental health, and sleep in office workers [19]. Participants were equipped with a thigh-worn activPAL3 (attached with waterproof adhesive film) and a waist-worn ActiGraph wGT3X-BT (worn on a belt during waking hours). Both sensors were worn on the right body side for an average of 7 days. Participants kept a diary to note sensor wear-time.

### 2.3. Pre-Processing

A detailed pre-processing description can be found in [16]. Briefly, the pre-processing consisted of three steps: (1) activPAL wear-time detection; (2) sensor synchronisation; and (3) ActiGraph wear-time detection. The activPAL wear-time detection ensured that only days with at least 500 steps, at least 12 h (without bedtime), and no more than 95% of the time spent in one posture were analysed [20]. Bedtime was excluded with an established activPAL algorithm [20] and, in the case of a late start (after 1:00) or an early end (before 4:30), visually inspected and adjusted using the diary information. The subsequent sensor synchronisation ensured that the raw sensor data matched in time. This step was required since there was an obvious delay of the ActiGraph compared to the activPAL, evident in the raw signal comparison. Finally, ActiGraph non-wear-time was excluded by removing all events with a constant ActiGraph raw signal for ≥90 min or if the activPAL recorded a posture change or classified part of the event as locomotion. The pre-processing finally led to synchronised raw and proprietary sensor data (activPAL event file, generated with activPAL3 v7.2.38; ActiGraph counts-per-second file, generated with ActiLife v6.13.4 using the low-frequency-extension filter) for the time both sensors were worn, hereinafter referred to as wear-time.

### 2.4. activPAL Algorithm Development

The development of the energy expenditure algorithm for the activPAL was split into an algorithm training to develop several algorithms with machine learning and an algorithm selection to identify the single best algorithm to be used subsequently in the method comparison (Figure 1). Only the data of the development sample (*n* = 38) were used.

For the algorithm training, all activPAL sitting events ≥ 1 min were identified and as many integer minutes as possible were extracted. For example, from a sitting event of 5.5 min, the middle 5 min were extracted. Each sitting minute was labelled as inactive or active using the synchronised ActiGraph counts-per-minute (cpm) and a cut-point of 75. In a previous study, the 75 cpm cut-point turned out to have a substantially higher validity (kappa of 0.69 compared to an indirect calorimeter) than the frequently used 100 cpm (kappa of 0.56) to separate inactive (≤1.5 METs) and active sitting (>1.5 METs) [21]. Subsequently, three different feature sets were generated: Feature Set 1 consisted of the same 563 signal features already used for the ActiGraph algorithm development [15]. Feature Set 2 consisted of the 213 time-based features for the raw sensor axis and vector magnitude of Feature Set 1. Feature Set 3 consisted of 52 cross-recurrent features calculated for the raw sensor axis and vector magnitude [22]. Note that each feature (e.g., range) was treated for each sensor axis (e.g., *x*-axis) independently. For Feature Sets 1 and 2, a random forest classifier limited the number of features to the most relevant 100 [23]. The features were then iteratively included into a bagged classification tree ensemble in MATLAB (Mathworks Inc., Natick, MA, USA). In each iteration, the feature with the highest cross-validity was added and the corresponding algorithm was retrained with optimised training properties (using the fitcensemble-function with hyperparameteroptimisation set to “all”). This technique searched for the optimal learning method (boosting or bagging), split criterion (gini diversity index, deviance, or twoing), tree size, maximum number of splits, minimum leaf size, and learning rate (all numerical). The iterative feature inclusion stopped when the maximum cross-validity was reached (no further increase for the next 10 features). The cross-validity was assessed with the mean of sensitivity and specificity using the leave-one-subject-out approach [14,24].

The subsequent algorithm selection identified the most accurate (lowest Bland–Altman bias) and precise (narrowest 95% limits of agreement) algorithm to measure sedentary time. Differently from the training data, the data for the algorithm selection included all activPAL sitting events of the development sample, regardless of duration. Thus, for the 5.5-min sitting example, the first 5 non-overlapping minutes were extracted beginning with the start of the event, and the last minute was extracted so it ended at the end of the sitting event. This procedure caused an overlap of the last and second last minute. However, the last minute was only used to classify the remaining fraction of the event (0.5 min in the 5.5-min example). To classify sitting events < 1 min, the input minute equally overlapped the start and end of the event, but again only the event itself was classified. For each algorithm, the sedentary estimate for each day was put in relation to wear-time, averaged over all days for each participant, and the bias and 95% limits of agreement were calculated by subtracting the sedentary estimate of POPAI. Only the most accurate and precise algorithm was subsequently used in the independent method comparison; the accuracy and precision, as well as the properties of all remaining algorithms, are presented in Appendix A.

### 2.5. Independent Method Comparison

Sedentary behaviour was classified with each method independently using the data of the comparison sample (*n* = 62).

The POPAI processing started with the activPAL event file and reclassified each sitting event on a minute-by-minute basis into inactive (equal to sedentary) and active sitting using the ActiGraph counts and a cut-point of 75 cpm [16]. Sitting events < 1 min and the remaining fraction of longer events were classified with the corresponding fraction of the cut-point (e.g., 25 counts for a 20-s sitting event). When the classification changed during a sitting event, the event was split accordingly into sedentary behaviour and active sitting.The activPAL processing directly used the activPAL event file and classified each sitting event as sedentary behaviour, just as a typical field study does [25].The activPAL+ processing started with the activPAL event file and reclassified each sitting event on a minute-by-minute basis into inactive (equal to sedentary) and active sitting using the new activPAL energy expenditure algorithm. Sitting events < 1 min and the remaining fraction of longer events were classified as for the selection of the best activPAL algorithm. When the classification changed during a sitting event, the event was split accordingly into sedentary behaviour and active sitting.The ActiGraph processing directly used the ActiGraph counts-per-second file and classified each minute with <100 counts as sedentary behaviour, just as a typical field study does [25].The ActiGraph+ processing used the new ActiGraph posture algorithm [15] and classified each minute into sitting, standing, and locomotion and subsequently reclassified each sitting minute into sedentary behaviour and active sitting using the corresponding ActiGraph counts and a cut-point of 75 cpm.

For each method, total sedentary time per day was averaged for each participant over all days with ≥10 h of recording. The method comparison used the bias as a measure of accuracy (group level) and the 95% limits of agreement as a measure of precision (individual level) [26]. Both were calculated by subtracting the POPAI sedentary estimate from the single sensor estimates (activPAL, activPAL+, ActiGraph, and ActiGraph+). In the case the bias and/or the 95% limits of agreement depended on the mean of both methods, the regression approach was used [26]. For better understanding, the accuracy was additionally expressed in relation to the sedentary estimate of POPAI (relative bias, which indicates the relative over-/underestimation), the precision was additionally expressed with the root mean square error, and the correlation to POPAI was expressed with the Pearson correlation coefficient. The comparison was done with absolute (minutes per day) and relative numbers (as a percentage of wear-time). However, only the absolute numbers are presented to simplify interpretation, while the relative numbers can be found in the Appendix A.

To analyse whether the accuracy (bias) and precision (95% limits of agreement) of the single sensors varied over the day, both were additionally calculated for non-overlapping 30-min intervals, e.g., from 12:00 to 12:30 or from 12:30 to 13:00, and expressed in relation to the interval length. The results were plotted against daytime, separately for weekdays and weekend days, but only for intervals with data from at least 80% of the participants, i.e., *n* ≥ 50.

Finally, the Bland–Altman analysis was repeated for sedentary time accumulated in bouts ≥10 min and ≥30 min since future studies might want to clarify the impact of prolonged sedentary bouts on health.

## 3. Results

The development sample spent 395 ± 72 min or 6.6 ± 1.2 h a day sedentary, which is equal to 43.9 ± 7.9% of the 14.9 ± 0.8 h wear-time per day (mean ± SD of POPAI).

The comparison sample spent 411 ± 80 min or 6.9 ± 1.3 h a day sedentary, which is equal to 45.7 ± 8.3% of the 14.9 ± 0.8 h wear-time a day (mean ± SD of POPAI). Furthermore, the comparison sample accumulated 242 ± 84 and 75 ± 53 min per day or 26.9 ± 9.1% and 8.3 ± 5.9% of the wear-time in bouts ≥10 and ≥30 min.

### 3.1. activPAL Algorithm Development

The most accurate and precise activPAL algorithm used 10 time- and 2 frequency-based features from Feature Set 1 (features are given in the Appendix A, Appendix A). The algorithm combines 466 decision trees trained with gentle adaptive boosting, a learning rate of 0.43, a minimum leaf size of 9, and a maximum of 1253 splits. The algorithm is freely available from GitHub (https://github.com/RomanKuster/POPAIv2.0, accessed on 26 May 2021). In the development sample, the algorithm’s bias (±standard error) was 6 ± 1 min per day, while the 95% limits of agreement ranged from −3 ± 1 to 14 ± 1 min per day.

### 3.2. Method Comparison

All single-sensor methods overestimated total sedentary time per day compared to POPAI (Table 1). The ActiGraph+ had the lowest bias (absolute and relative), while the activPAL+ had the narrowest 95% limits of agreement and highest correlation. The 95% limits of agreement for both the activPAL+ and ActiGraph+ were lower than the biases of their proprietary counterparts (Figure 2).

The analysis in relation to daytime showed for both new methods (activPAL+ and ActiGraph+) a lower bias and narrower 95% limits of agreement compared to their proprietary counterparts (activPAL, ActiGraph) for weekdays and weekend days (Figure 3). On weekdays, the activPAL+ bias and 95% limits of agreement remained roughly constant over the entire day, while the ActiGraph+ bias was slightly higher and the 95% limits of agreement slightly wider in between 8:00 and 16:00 compared to after 17:00. Both observations reflect the bias and 95% limits of agreement of the corresponding proprietary data processing in an attenuated manner. No such pattern was observed on weekend days.

For sedentary time accumulated in bouts ≥10 and ≥30 min, the ActiGraph+ was the only method not overestimating but underestimating sedentary time (negative bias, Table 2). The ActiGraph+ had the lowest bias (absolute and relative) and root mean square error, as well as the narrowest 95% limits of agreement and highest correlation. As for total sedentary time, the 95% limits of agreement of both the activPAL+ and ActiGraph+ were lower than the biases of their proprietary counterparts.

## 4. Discussion

The present study analysed the accuracy (bias) and precision (95% limits of agreement) of measuring both components of sedentary behaviour, posture (i.e., sitting, reclining, or lying) and energy expenditure (i.e., ≤1.5 METs), with a single sensor: a thigh-worn activPAL or a waist-worn ActiGraph. All single-sensor methods overestimated total sedentary time per day compared to POPAI. Thus, the two-sensor POPAI should remain the first choice to measure sedentary behaviour compliant with its definition. However, if the use of two sensors is deemed unsuitable—which will be most likely the rule rather than the exception—our recommendation is to combine the proprietary data processing of the single sensors with the corresponding new algorithm: the energy expenditure algorithm when using the activPAL and the posture algorithm when using the ActiGraph.

Both algorithms substantially reduced the overestimation of total sedentary time: from 129 to 21 min per day for the activPAL and from 84 to 7 min per day for the ActiGraph (Table 1). In relation to the sedentary time measured by POPAI, adding the activPAL energy expenditure algorithm reduced the overestimation from 33.2% to 5.4%, while adding the ActiGraph posture algorithm reduced the overestimation from 22.1% to 1.9%. Whether the rather small overestimation and thus the bias of the activPAL+ and ActiGraph+ is of any relevance with respect to health needs to be clarified in future studies. In the case it is irrelevant, the activPAL+ and the ActiGraph+ can be considered interchangeable with POPAI at the group level. In the case it is relevant, one should adjust for the bias by subtracting it from the sedentary estimate of each participant [26], which would make the methods interchangeable at the group level. Adjusting for the bias would centre the 95% limits of agreement around zero (while keeping its range), reduce the root mean square error of the activPAL from 26 to 16 min per day, and reduce the root mean square error of the ActiGraph from 26 to 25 min per day.

Another, probably even more important, figure of the method comparison is the 95% limits of agreement for which there is no straightforward adjustment available. The 95% limits of agreement cover the range within which 95% of the differences to POPAI will lie at the individual level. In the case the 95% limits of agreement cover a range that can be considered irrelevant, one could use the two methods interchangeably at the individual level [26]. Here, the 95% limits of agreement for the activPAL+ ranged from −10 to +52 min per day and the 95% limits of agreement for the ActiGraph+ ranged from −43 to +57 min per day. Even though the range was considerably smaller than for the proprietary data processing (activPAL: +50 to +208 min per day; ActiGraph: +16 to +151 min per day), a range of ≥62 min per day might still be relevant. Accordingly, the activPAL+ and the ActiGraph+ cannot be considered interchangeably with POPAI at the individual level, supporting our conclusion that POPAI should remain the first choice to measure sedentary behaviour compliant with its definition.

Even though they are not interchangeable, one must bear in mind that the 95% limits of agreement of both the activPAL+ and the ActiGraph+ did not span over the bias of their proprietary counterparts. This indicates that the precision with the new algorithms at the individual level (95% limits of agreement) is higher than the accuracy without the new algorithms at the group level (bias). Furthermore, adding the new algorithms resulted in rather low root mean square errors (26 min per day) and an almost perfect correlation to POPAI (0.98 and 0.95, respectively). Both the activPAL+ and the ActiGraph+ might therefore be reasonably accurate and precise choices with the benefit that only one sensor needs to be used. Moreover, research groups already using one of the two sensors can apply the algorithms without having to spend money on new hardware to improve the accuracy and precision of their sedentary measurements substantially, even retrospectively on data already collected. Both new methods are freely available on GitHub (https://github.com/RomanKuster/POPAIv2.0, accessed on 26 May 2021), and researchers without access to MATLAB are welcome to contact the corresponding author to process the data with activPAL+ and/or ActiGraph+.

The analysis in relation to daytime showed a rather constant accuracy (bias) and precision (95% limits of agreement) for the activPAL and activPAL+, and a somewhat lower accuracy and precision for the ActiGraph and ActiGraph+ from 08:00 to 16:00 compared to after 17:00 on weekdays. Since all included participants were office workers, it seems that the activPAL+ might be the better choice for studies focusing on office work (the domain in which our participants spent most of their time from 08:00 to 16:00 on weekdays), while the sensor choice has less bearing for the time after 17:00 on weekdays and for weekend days. This observation confirms the results of a previous laboratory study in which we noticed the highest accuracy for lower-body sensors to measure both components of sedentary behaviour in desk based office work [27] and adds the information that this does not directly translate to outside the office. In this regard, it is important to note that the bias of the activPAL perfectly reflects active sitting (which was spread throughout the day), while the bias of the ActiGraph almost perfectly reflects inactive standing (which was most dominant between 08:00 and 16:00) [16]. In other words, the investigated office workers sat equally active throughout the weekdays but stood more inactive during office hours than afterwards.

The current evidence on whether the time spent in prolonged sedentary bouts matters is inconclusive, and future studies might want to clarify this issue [28]. Therefore, the present study investigated the accuracy and precision of the single sensors to measure sedentary time accumulated in bouts ≥10 and ≥30 min. The results demonstrate that the proprietary processing of the single sensors substantially overestimates sedentary time: by ≥100% and ≥500% for the activPAL and by 25% and 50% for the ActiGraph (Table 2). Accordingly, due to the serious lack of accuracy and precision of the single sensors, we recommend reanalysing the existing evidence on prolonged sedentary behaviour measured with the ActiGraph (e.g., [29]) and activPAL (e.g., [30]) using the new algorithms. The addition of the new algorithms substantially reduced the overestimation: down to 13.3% and 26.6% for the activPAL and down to −4.3% and −2.2% for the ActiGraph (which represents an underestimation). In this line, the addition of the new algorithms substantially reduced the range covered by the 95% limits of agreement for both sensors, which did not even span over the bias of the proprietary data processing (Table 2). This indicates that the precision with the new algorithms on an individual level (95% limits of agreement) is higher than the accuracy without the new algorithms on group level (bias). The ActiGraph+ had the highest accuracy and precision and should therefore be the single-sensor method of choice for the sedentary bout analysis.

### Critical Appraisal

The present study is subject to some critical aspects that should be carefully considered. First, POPAI, our reference method, is a new method, which questions its use as a reference method here. However, POPAI combines the well-established proprietary data processing of the activPAL to classify posture with the well-established proprietary data processing of the ActiGraph to classify energy expenditure [16]. Compared to the standard processing of the two sensors, the most serious difference is the use of a lower cut-point to separate inactive and active sitting (75 cpm instead of 100 cpm). We compared the cut-point validity in a preceding study to an indirect calorimeter and observed that the 100 cpm is a fairly valid choice when separating inactive and active behaviours without consideration of posture, but the 75 cpm is a much more valid choice when separating only inactive and active sitting [21]. Accordingly, and in line with the literature, the ActiGraph processing without consideration of posture used the 100 cpm cut-point [3,10], while the POPAI and ActiGraph+ processing with consideration of posture used the 75 cpm cut-point [21]. Combining the validity of the activPAL posture classification [8,31,32] with the validity of the 75 cpm cut-point [21,33] actually leads to an estimated sensitivity and specificity of 92.5% and 91.9% to measure sedentary behaviour with POPAI (see Appendix A). Since we are not aware of any similarly valid method using established sensors to measure free-living sedentary time, we consider the use of POPAI as a reference method to be justified and in fact a strength of this study. An alternative would have been to use a two-sensor Vitaport or VitaMove system [17,34]. However, the validity of these sensor combinations is unknown to us, and our participants would then have had to wear four sensors.

Another limitation of this study is the lack of predefined satisfactory accuracy and precision limits [26]. Based on the available evidence collected with movement sensors [3], we did not feel qualified to quantify meaningful and relevant accuracy and precision limits. The lack of an absolute limit, however, does not mean that one method could not be more accurate (when having a lower bias) and precise (when having narrower 95% limits of agreement) than another, nor that a particular method cannot be inaccurate and imprecise. In fact, whether a method overestimating sedentary time by 84 min per day with 95% limits of agreement covering a range of 167 min per day is accurate and precise is highly questionable. However, the evidence of the adverse health effects of sedentary behaviour has almost exclusively been collected with such a method: the 100 cpm cut-point for the ActiGraph [3].

The next limitation that needs special attention is the reuse of data captured for another purpose and the resulting sample size. We did not perform a sample-size calculation to develop the algorithms or compare the methods. To include a development sample of 38 office workers was a convenient choice taken for the ActiGraph algorithm development and kept for the activPAL algorithm development for consistency. The typical algorithm development in free-living uses approximately half the size of our sample [24], which is why the development sample should be seen as rather large. The size of the comparison sample was then a consequence. We strictly separated the two samples to ensure an independent method comparison, and only the data of 62 office workers remained. Consequently, we did not perform any statistical inference testing. We described the differences between the methods explanatively and added wherever possible a measure of certainty (i.e., standard error and 95% confidence intervals). We are aware that standard errors and confidence intervals could be used to detect significant differences (which here would lead to rather low p-values, i.e., <0.001) [35], but we see no benefit in doing so when analysing accuracy and precision [26]. For those researchers less familiar with Bland-Altman statistics, we additionally included the relative bias (indicating over-/underestimation), the root mean square error, and the Pearson correlation as additional measures to compare the methods. In this light, we consider the reuse of data captured for another purpose as an additional strength of this study. All participants were recorded before starting this project, which means that the researchers in charge of data recording were completely blinded to the purpose of this study, and the sensors were used as in a normal field study. This includes practical aspects of data recording that likely reduced the accuracy and precision of the developed algorithms, such as sensor non-wear or sensor worn upside-down. The sample used for the method comparison spent 9.0 ± 1.4 h per day sitting (activPAL data) and 8.3 ± 1.2 h per day inactive (ActiGraph data). This is in the range previously described by others [25,36,37,38], and the presented accuracy and precision might very well reflect the sensor use in future field studies. However, this study was limited to office workers. From the daytime analysis for weekdays and weekend days shown in Figure 3, there is no evidence that using the algorithms will lead to less accurate and precise sedentary measurements in other populations, but future studies are needed to verify this.

After completing the method comparison, we additionally investigated the performance of the activPAL+ if we had chosen another algorithm to inspect our algorithm selection (see Appendix A). This analysis showed that the chosen algorithm also had the highest accuracy and was among the algorithms with the highest precision in the comparison sample. We interpret this as a justification of our algorithm selection procedure, which identified the single best activPAL algorithm to measure sedentary behaviour. Moreover, the comparison among all algorithms for each feature set (Appendix A) showed that no feature set generalised markedly better than another did, a finding that stands in contrast to those of Montoye et al., who noticed a better generalisability for time- than frequency-domain features for a wrist-worn sensor [39]. However, only decision trees were trained in this study, and it remains unknown how the feature sets perform with other machine learning techniques such as k-nearest neighbours or support vector machines. Since we have no evidence that there is a serious difference between different machine learning techniques [24,40], we decided to limit this study to decision trees and trained the trees with optimised parameters (e.g., allowing for boosting and bagging) and different feature sets to improve the classification.

## 5. Conclusions

The present study showed that measuring sedentary behaviour compliant with its definition is inaccurate and imprecise when using the proprietary data processing of the activPAL or the ActiGraph. A much more accurate and precise single-sensor method is to combine the proprietary data processing with an energy expenditure algorithm (activPAL+) or a posture algorithm (ActiGraph+). In fact, the precision at the individual level was higher when adding the algorithms than the accuracy at the group level when not adding the algorithms. The ActiGraph+ was the most accurate method showing its strength especially in the assessment of prolonged sedentary behaviour, while the activPAL+ was the most precise method showing its strength especially during office hours. Even though neither was interchangeable with the two-sensor POPAI, the activPAL+ and the ActiGraph+ are reasonably accurate and precise choices to measure sedentary behaviour in applied surveillance and research settings with a single sensor, and we recommend using them to improve our understanding of sedentary behaviour and its health effects. Both methods rely only on the raw and proprietary sensor data and can even be retrospectively applied on data already captured. The methods are freely available from GitHub (https://github.com/RomanKuster/POPAIv2.0, accessed on 26 May 2021), and researchers without access to MATLAB are welcome to contact the corresponding author for data processing.

## Figures and Tables

**Figure 1 ijerph-18-05782-f001:**
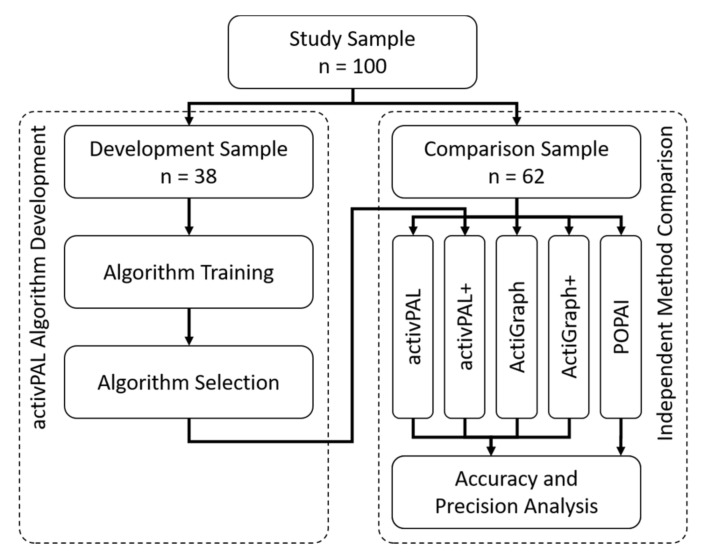
Flow-chart of the study with the development of the new activPAL energy expenditure algorithm for activPAL+ (**left**) and the subsequent independent method comparison with accuracy and precision (**right**).

**Figure 2 ijerph-18-05782-f002:**
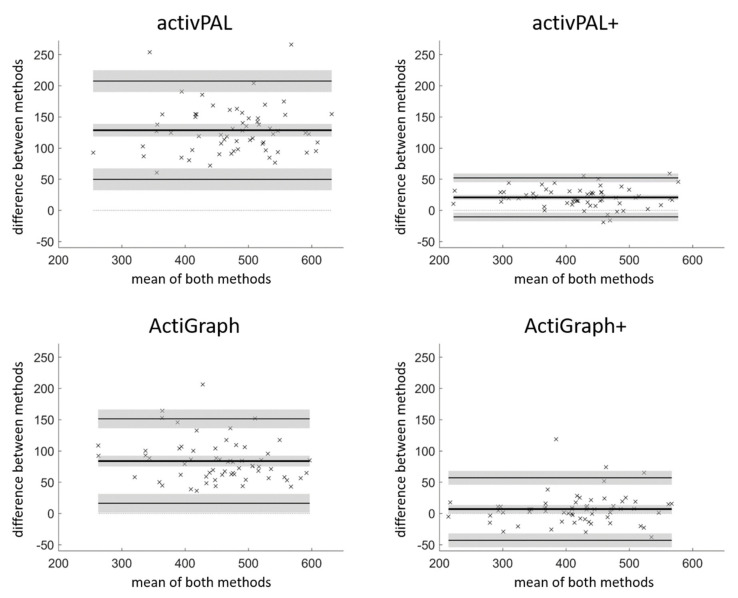
Bland–Altman plots for average sedentary time per day, expressed in minutes per day. The bias (bold line) and 95% limits of agreement (thin lines) are indicated, both with 95% confidence intervals (in grey). The corresponding figure in per cent of wear-time is given in the Appendix A (Appendix A).

**Figure 3 ijerph-18-05782-f003:**
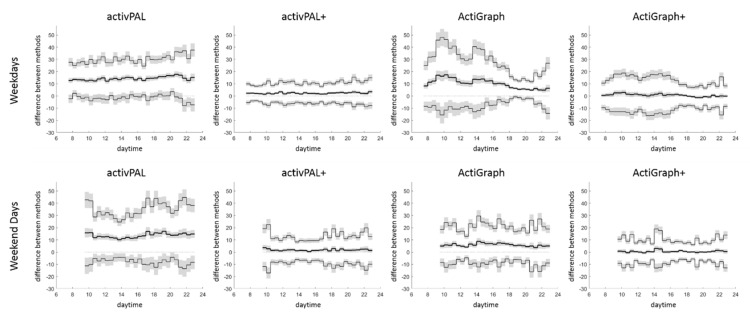
Bias (bold line) and 95% limits of agreement (thin lines) for total sedentary time over 30-min intervals throughout weekdays and weekend days, with 95% confidence intervals (in grey). The data are presented as per cent of the interval length (100% equals 30 min) and shown for intervals with valid wear-time of ≥50 participants, i.e., 07:30–23:00 for weekdays and 09:30–23:00 for weekend days.

**Table 1 ijerph-18-05782-t001:** Average sedentary time per day measured with each single-sensor method compared to POPAI.

Method	Sedentary Time (±SD)	Absolute Bias (±SE)	Relative Bias (±SE)	[95% LoA] (±SE)	RMSE	Correlation [CI]
activPAL	540 ± 82	129 ± 5	33.2 ± 2.0%	[50; 208] ± 9	134	0.88 [0.81–0.93]
activPAL+	432 ± 79	21 ± 2	5.4 ± 0.5%	[−10; 52] ± 3	26	0.98 [0.97–0.99]
ActiGraph	495 ± 73	84 ± 4	22.1 ± 1.6%	[16; 151] ± 7	90	0.91 [0.85–0.94]
ActiGraph+	418 ± 83	7 ± 3	1.9 ± 0.8%	[−43; 57] ± 6	26	0.95 [0.92–0.97]

Data are given as minutes per day, except relative bias (in per cent of POPAI sedentary time and thus indicating over-/underestimation) and correlation (unitless). Abbreviations: SD, Standard Deviation; SE, Standard Error; LoA, Limits of Agreement; RMSE, Root Mean Square Error; CI, Confidence Interval. The corresponding table in per cent of wear-time is given in the Appendix A (Appendix A).

**Table 2 ijerph-18-05782-t002:** Average sedentary time per day accumulated in bouts ≥10 and ≥30 min measured with each single-sensor method compared to POPAI.

Method	Sedentary Time (±SD)	Absolute Bias (±SE)	Relative Bias (±SE)	[95% LoA] (±SE)	RMSE	Correlation [CI]
Sedentary time accumulated in bouts ≥10 min						
activPAL	447 ± 83	205 ± 8	105.3 ± 10.1%	[74; 336] ± 14	215	0.69 [0.54–0.80]
activPAL+	268 ± 87	26 ± 4	13.3 ± 2.2%	[−38; 90] ± 7	41	0.93 [0.89–0.96]
ActiGraph	295 ± 87	53 ± 3	25.3 ± 2.2%	[2; 103] ± 6	58	0.96 [0.93–0.97]
ActiGraph+	233 ± 87	−9 ± 2	−4.3 ± 1.0%	[−47; 29] ± 4	21	0.98 [0.96–0.99]
Sedentary time accumulated in bouts ≥30 min						
activPAL	263 ± 73	188 ± 8 ^#^	531.3 ± 115.9%	[68; 309] ± 13	199	0.51 [0.30–0.68]
activPAL+	84 ± 54	10 ± 3	26.6 ± 7.5%	[−45; 64] ± 6	29	0.87 [0.80–0.92]
ActiGraph	100 ± 61	25 ± 2 ^#^	49.8 ± 6.4%	[−11; 61] ± 4	32	0.95 [0.92–0.97]
ActiGraph+	73 ± 55	−2 ± 1	−2.2 ± 3.0%	[−22; 19] ± 2	10	0.98 [0.97–0.99]

Data are given as minutes per day, except relative bias (in per cent of POPAI sedentary time and thus indicating over-/underestimation) and correlation (unitless). The hashtags mark the methods for which the bias depended on the mean of both methods. Abbreviations: SD, Standard Deviation; SE, Standard Error; LoA, Limits of Agreement; RMSE, Root Mean Square Error; CI, Confidence Interval. The corresponding table in per cent of wear-time is given in the Appendix A (Appendix A). The hashtags (#) mark the methods for which the bias depended on the mean of both methods.

## Data Availability

The data presented in this study are available on reasonable request from the last author (Ö.E.). The data are not publicly available due to ethical restrictions.

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
