# Peer review of "How Accurate and Precise Can We Measure the Posture and the Energy Expenditure Component of Sedentary Behaviour with One Sensor?"

_ijerph, 2021, doi:10.3390/ijerph18115782_

Round 1

Reviewer 1 Report

This study investigated the accuracy and precision of measurements for sedentary behavior, posture, and physical activity with a single sensor. They showed that combining proprietary data with an algorithm is an easy way to increase the accuracy and precision of single sensor sedentary estimation.

1. In order to easily understand the research design, please present the study flow chart as a figure including activPAL algorithm development and independent method comparison.

2. How accurately do the three different feature sets predict "inactive" defined by ActiGraph?

3. What is the sensitivity and specificity of sedentary measurements (or “inactive”) by various machine learning algorithms (ex. k-Nearest Neighbor, Support Vector Machine, Random Forest, Decision Tree, et al)?

4. It is necessary to compare machine learning predictive models of sedentary measurements (or "inactive"). Please present ROC curves for the models used in the activPAL algorithm development. Also, are there any statistically significant differences between the ROC curves?

Author Response

Point 1. In order to easily understand the research design, please present the study flow chart as a figure including activPAL algorithm development and independent method comparison.

Answer 1. Study Flow Chart included as Figure 1, line 111 on page 3

Point 2. How accurately do the three different feature sets predict "inactive" defined by ActiGraph?

Answer 2. The average accuracy (bias) of feature set 1 is 2.4 ± 0.6 % of wear-time, the accuracy of feature set 2 is 2.5 ± 0.3% of wear-time, and the accuracy of feature set 3 is 4.8 ± 2.3% (see supplementary material, Table S1, “Average (mean ± SD)”). We added a statement on line 181-183, page 4: “… the accuracy and precision as well as the properties of all remaining algorithms can be found in supplementary material, Table S1 and S2.”. We also added a statement on line 449-450, page 11: “… the comparison among all algorithms for each feature set (supplementary material, Table S1) showed …”

Point 3. What is the sensitivity and specificity of sedentary measurements (or “inactive”) by various machine learning algorithms (ex. k-Nearest Neighbor, Support Vector Machine, Random Forest, Decision Tree, et al)?

Answer 3. We limited this study to decision trees, trained with boosting and bagging techniques (which includes but is not limited to random forest). Statement added in method section on line 161-163 on page 4: “This technique searched for the optimal learning method (boosting or bagging), split criterion…”. Furthermore, we also added a section in the discussion on line 453-458 on page 11-12: “However, only decision trees were trained in this study and it remains unknown how the feature sets perform with other machine learning techniques such as k-nearest neighbours or support vector machines. Since we have no evidence that there is a serious difference between different machine learning techniques [24,40], we decided to limit this study to decision trees and trained the trees with optimized parameters (e.g. allowing for boosting and bagging) and different feature sets to improve the classification.”

Point 4. It is necessary to compare machine learning predictive models of sedentary measurements (or "inactive"). Please present ROC curves for the models used in the activPAL algorithm development. Also, are there any statistically significant differences between the ROC curves?

Answer 4. We fully agree with the reviewer that ROC curves are a useful and informative way to compare different machine-learning algorithms. In this study, however, we used Bland-Altman statistics to select the algorithm, which is why we believe that ROC curves would confuse the reader. We selected the algorithm with lowest bias and narrowest 95% limits of agreement, which is not necessarily the one with the best ROC curve (e.g. largest area under the ROC curve). Accordingly, we hope that the reviewer agrees that for this particular study, even though informative, ROC curves will be rather confusing than clarifying. Furthermore, following the work of Bland and Altman, it is much more informative to present bias and 95% limits of agreement instead of significant effects. This is why we have not performed any statistical inference testing, but present the method comparison explanatively, and added wherever possible a measure of certainty (see discussion, line 419-426 on page 11).

Reviewer 2 Report

The work is of great interest. Sedentary behavioris an emergent public health topic, but there is still no method to simultaneously measure both components of sedentary behavior: posture and physical activity. Authors recently presented a simultaneous measurement of both components of sedentary behavior: the Posture and Physical Activity Index (POPAI). However, the most serious limitation of POPAI is the need for two sensors, significantly limiting its suitability for applied surveillance and research settings. Therefore, the present study examined the accuracy and precision of measuring both components of sedentary behavior with one single sensor.

The methodological technique used can be used in the practice of researching the effects of a sedentary lifestyle.

The work is well illustrated and clearly reflects the findings.

Author Response

Thank you very much for your positive feedback.

Reviewer 3 Report

This work investigates the accuracy and precision to measure sedentary time when combining the activPAL posture sensor with a new activity algorithm, and the ActiGraph activity sensor with a previously published posture algorithm. One hundred office workers wore both sensors for an average of 7 days. The single sensor sedentary estimates were compared with Bland-Atman statistics, and a combined measurement with both sensors. Results confirm that adding the algorithms reduced the overestimation from 129 to 21 (activPAL) and from 84 to 7 minutes a day (ActiGraph), with 95% limits of agreements.

The results are relevant for the knowledge area, and conclusions are supported by the results. Nevertheless, some important information should be included before this manuscript could be considered for possible publication.

The activPAL algorithm is not properly presented.

The algorithm training should be formally defined.

The tested algorithms with machine learning should be mentioned, and the selected machine learning algorithm should be presented (including all relevant parameters).

Author Response

Point 1. The activPAL algorithm is not properly presented.

Answer 1. We added a more detailed description of the selected algorithm in the result section (line 242-246 on page 6): “… (features given in supplementary material, Table S1). The algorithm combines 466 decision trees trained with gentle adaptive boosting, a learning rate of 0.43, a minimum leave size of 9 and a maximum of 1’253 splits. The algorithm is freely available from GitHub (published alongside this manuscript, referenced with www address here).” This gives the interested reader direct access to the selected algorithm.

Point 2. The algorithm training should be formally defined.

Answer 2. We added a more detailed description of how we trained the algorithms on line 161-163 on page 4: “This technique searched for the optimal learning method (boosting or bagging), split criterion (gini diversity index, deviance, or twoing), tree size, maximum number of splits, minimum leaf size, and learning rate (all numerical).”

Point 3. The tested algorithms with machine learning should be mentioned, and the selected machine learning algorithm should be presented (including all relevant parameters).

Answer 3. In addition to describing the selected algorithm and the training in more detail, we also added a new table in the supplemental material showing the properties of all developed algorithms (Table S2), and reference the table in the method section line 181-183 on page 4: “…the accuracy and precision as well as the properties of all remaining algorithms can be found in supplementary material, Table S1 and S2.”.

Round 2

Reviewer 3 Report

The recommendation were addressed by the authors, the manuscript could be consider for publication.

Author Response

We'd like to thank you once again for helping us to improve the quality of our manuscript.